# Diagnostic Algorithm for Surgical Management of Limbal Stem Cell Deficiency

**DOI:** 10.3390/diagnostics13020199

**Published:** 2023-01-05

**Authors:** Boris Malyugin, Svetlana Kalinnikova, Ruslan Isabekov, Dmitriy Ostrovskiy, Boris Knyazer, Maxim Gerasimov

**Affiliations:** 1S. Fyodorov Eye Microsurgery Federal Institution, 127486 Moscow, Russia; 2Department of Ophthalmology, A. Yevdokimov Moscow University of Medicine and Dentistry, 127486 Moscow, Russia; 3Department of Ophthalmology, Soroka University Medical Center and Faculty of Health Sciences, Ben-Gurion University of the Negev, Beer-Sheva 8410501, Israel

**Keywords:** limbal stem cell deficiency, fluorescein staining, impression cytology, anterior segment optical coherence tomography, in vivo confocal microscopy, ocular surface reconstruction, simple limbal epithelial transplantation, glueless simple limbal epithelial transplantation, labial mucosal epithelium grafting, simple oral mucosa epithelial transplantation, paralimbal oral mucosa epithelium transplantation

## Abstract

Background: Limbal stem cell deficiency (LCSD) presents several challenges. Currently, there is no clearly defined systematic approach to LSCD diagnosis that may guide surgical tactics. Methods: The medical records of 34 patients with LSCD were analyzed. Diagnostic modalities included standard (visometry, tonometry, visual field testing, slit-lamp biomicroscopy with corneal fluorescein staining, Schirmer test 1, ultrasonography) and advanced ophthalmic examination methods such as anterior segment optical coherence tomography, in vivo confocal microscopy, impression cytology, and enzyme-linked immunoassay. Results: Standard ophthalmological examination was sufficient to establish the diagnosis of LSCD in 20 (58.8%) cases, whereas advanced evaluation was needed in 14 (41.2%) cases. Depending on the results, patients with unilateral LSCD were scheduled to undergo glueless simple limbal epithelial transplantation (G-SLET) or simultaneous G-SLET and lamellar keratoplasty. Patients with bilateral LSCD with normal or increased corneal thickness were enrolled in the paralimbal oral mucosa epithelium transplantation (pLOMET) clinical trial. Conclusions: Based on the diagnostic and surgical data analyzed, the key points in LSCD diagnosis were identified, helping to guide the surgeon in selecting the appropriate surgical procedure. Finally, we proposed a novel step-by-step diagnostic algorithm and original surgical guidelines for the treatment of patients with LSCD.

## 1. Introduction

Limbal stem cell deficiency (LSCD) is an ocular surface disease caused by a decrease in the population and/or function of corneal epithelial stem/progenitor cells, according to the global consensus on definition classification, diagnosis, and staging [1]. According to the X-Y-Z theory, a disturbance in the homeostasis of the corneal epithelium leads to a shift in the equilibrium described by Thoft [2]. When component X (proliferation of basal corneal epithelial cells) is decreased, there are insufficient cells to balance component Z (normal corneal epithelial loss). Thus, the LSCD-affected eye cannot have a stable corneal epithelium despite normal or increased component Y (proliferation and centripetal migration of the amplifying cells). This leads to the breakdown of the corneal-conjunctival barrier and invasion of the conjunctiva into the cornea [1]. LSCD could be an independent acquired entity or a result of various systemic or genetic diseases. Among them, ocular burns and contact lens wear represent a group of acquired non-immune-mediated LSCD [1]. In cases of Stevens-Johnson syndrome (SJS), mucous membrane pemphigoid (MMP), etc., the ocular surface is a target tissue [3,4], and LSCD is classified as primary immune-mediated [1]. 

Clinically, LSCD manifests as partial or total corneal conjunctivalization, and neovascularization occurs as a consequence of the breakdown of the corneal-conjunctival barrier [5]. LSCD is often accompanied by a substantial decrease in vision and chronic eye inflammation and may progress to severe conditions involving recurrent corneal erosions, corneal thinning, and perforation [6]. The management of LSCD depends on various factors. Recently, it was suggested to optimize the ocular surface by topical treatment and surgical reconstruction of eyelid anomalies and symblepharon. This is followed by surgical interventions aimed at corneal epithelium reconstruction [7]. Conjunctival limbal autograft (CLAU) [8], cultivated limbal epithelial transplantation (CLET) [9], and simple limbal epithelial transplantation (SLET) [10] are well-established procedures for corneal epithelium reconstruction in patients with unilateral LSCD, with predictable high anatomical success rates [11]. In patients with bilateral LSCD, different methods of corneal re-epithelialization [12,13] and keratoprosthesis surgery [14,15] could be considered. 

The selection of a proper surgical procedure in LSCD patients mostly relies on the correct diagnosis, surgeon preferences, and available technologies. Wettability of the ocular surface, disease progression rate, laterality, the main cause of the disease, and the stage of LSCD are all very important factors in decision-making, allowing the establishment of the best strategy [16]. Currently, LSCD can be diagnosed using various methods [17]. Slit-lamp biomicroscopy with fluorescent staining is the most useful and widely used approach for this purpose. More comprehensive methods include impression cytology (IC), anterior segment optical coherence tomography (AS-OCT), and in vivo laser scanning confocal microscopy (IVCM). 

IC is used to diagnose various ocular surface diseases, including LSCD [18]. IC detects the presence of conjunctival and/or goblet cells in the cornea, confirming the migration of these cells through the damaged corneal limbal barrier. IC may provide an in-depth analysis of biological markers in samples obtained from the LCSD-affected eye. For example, the expression of corneal and conjunctival keratins (K) and goblet cell mucin (MUC5AC) was demonstrated in patients with LSCD [19].

AS-OCT is used in patients with LSCD for advanced noncontact visualization of the anterior segment structures. AS-OCT can demonstrate corneal conjunctivalization, area, and depth of corneal stromal scarring, as well as structural changes in the corneal limbus and palisades of Vogt [17,19]. AS-OCT also provides an advanced evaluation of the corneal epithelial layer. The central corneal zone can be directly measured using standard and wide epithelial mapping. The thickness of the corneal limbus epithelium and fibrovascular pannus can also be defined with high precision. Finally, AS-OCT angiography (AS-OCTA) can be used to visualize the limbal vasculature and detect deep and superficial corneal neovascularization, both during initial diagnosis and documenting disease progression [17].

IVCM is a non-invasive diagnostic method for the cornea, limbus, and conjunctiva at the cellular level [17]. It was reported to serve as an advanced diagnostic and monitoring tools in patients with LSCD [20,21]. In vivo IVCM scans have a very high resolution and help analyze healthy and pathologically affected corneal limbus and the palisades of Vogt, morphological changes of the corneal epithelial and stromal cells, and reduction in the density of the sub-basal nerve plexus [17].

Diagnosis of LSCD can be easily defined in cases where the affected cornea is fully covered with vascular pannus or corneal conjunctivalization is documented during slit-lamp biomicroscopy with fluorescent staining. However, in patients with subtle clinical appearances, the diagnosis of LSCD may become challenging [22]. It is important to note that correct surgical tactics are vital for the anatomical and functional success of further procedures. Thus, the physicians’ final decision to select an appropriate surgical or therapeutic approach might be very challenging. In this study, we analyzed various diagnostic methods used in patients with LSCD and identified several critical decision-making parameters that can guide surgeons in selecting the appropriate surgical technique for ocular surface reconstruction. We also propose a novel step-by-step diagnostic algorithm and original surgical guidelines for the treatment of patients with LSCD, based on our clinical experience.

## 2. Materials and Methods

### 2.1. Medical Records Review

We reviewed the medical records of 34 patients with LSCD who underwent ocular surface reconstruction at the Fyodorov Eye Microsurgery Federal State Institution between 2017 and 2022. Data extracted for case-by-case analysis included anterior segment slit-lamp biomicroscopy with corneal staining, AS-OCT, AS-OCTA data, IVCM, IC, cytokine levels in tear samples, and type of surgery.

### 2.2. Ocular Surface Slit-Lamp Biomicroscopy and Staining

Ocular surface slit-lamp biomicroscopy and staining were performed according to the standard clinical protocol. Briefly, after biomicroscopy without staining, a sterile test strip with low-molecular-weight fluorescein (FluoStrips^®^, Contacare Ophthalmic and Diagnostics, Gujarat, India) was moisturized with one drop of antiseptic or artificial tear solution and applied to the lower conjunctival fornix. Next, the biomicroscopy was performed under cobalt blue light and yellow filters (BQ 900, Haag-Streit, Bern, Switzerland), and the appearance of the LSCD was photo-documented. Classification and staging of the LSCD were done according to the global consensus [1].

### 2.3. AS-OCT and AS-OCTA

The scans were performed using an Avanti xR optical coherence tomography device (Optovue Inc. Fremont, CA, USA) following manufacturer instructions in the “AS-OCT” modes using the Line, Cross-Line with a scan length of up to 8 mm and an axial resolution of 5 microns, Pachymetry, and PachymetryWide. Manual measurements of thicknesses were done in Line mode using the proprietary Avanti xR OCT program for corneal stromal and fibrovascular pannus of the affected cornea. Evaluation of the palisades of Vogt was performed on the upper limbus of the healthy eye according to the study [23]. The “Angio OCT” modes using the AngioVue program (software version SW 2018.1.1.63) and SSADA algorithm. Scanning of the central part of the cornea and limbal region in the Angio OCT mode was performed without the use of the corneal module. An AngioRetina scan was used with a 3 × 3 mm scan area on the retina. The scan resolution is 304 × 304 pixels. Scanning was carried out with the automatic settings turned off, in manual mode. The “Angio OCT” allows to observe uneducated vessels in a certain area of the cornea, objective monitoring of the progression and regression of vascularization before and after treatment.

### 2.4. In Vivo Confocal Microscopy (IVCM)

IVCM was performed using a laser confocal microscope HRT-II with a Rostock Cornea Module (Heidelberg Engineering GmbH, Germany) in the manual contact mode following the protocol described earlier [24]. Before the examination, the patient is instilled one drop of anesthetic to the lower conjunctival fornix. Then a drop of immersion liquid is placed on the lens of a confocal microscope, brought to the cornea before touching and set so that the thickness of the immersion liquid layer is 2 mm, then the intersection zone of interest is scanned. The LSCD-affected cornea was examined in 5 points: upper and lower limbus, central cornea, and nasal and temporal limbus. The size of the examination zone was 400 µm × 400 µm.

### 2.5. Impression Cytology and Immunofluorescent Staining (IC-IF)

IC-IF was done using a modified protocol. Samples were collected from patients with LSCD after topical anesthesia using the methylcellulose membrane from the sterile syringe filter (∅—13 mm, pore size—0.45 micron, FMC401013, Jet Bio-Filtration Co., Ltd., Guangzhou, China) was applied to the central and peripheral corneal areas of the LSCD-affected eye. Next, the samples were fixed in a 96% ethanol solution for 24 h at 4 °C. After that, the samples were incubated in PBS solution three times for 5 min at 24 °C, permeabilized with 0.3% Triton X-100 (A4975, AppliChem) in phosphate-buffered solution (PBST) for 15 min, and blocked with blocking solution (1% bovine serum albumin (LY-0081.050, Paneco) with 0.1% Twin 20 (A4974, AppliChem) in PBS) for 1 h. Afterward, the Samples were then stained with primary antibodies diluted in blocking solution against keratin 7 (K7, clone OV-TL 12/30, 1:100, ab216016, mouse monoclonal, Abcam, UK; keratin 12 (K12, 1:100, ab185627, rabbit monoclonal, Abcam), and incubated for 2 h at 37 °C in a humidified chamber. Next, the samples were washed in blocking solution three times for 5 min each and then incubated with paired secondary fluorescent antibodies (Alexa Fluor^®^488 (1:250, ab150077, goat anti-rabbit, Abcam) or Alexa Fluor^®^594 (1:250, ab150116, goat anti-mouse, Abcam) diluted in blocking solution in a humidified chamber for 1 h at 24 °C in the dark. Next, samples were washed in blocking solution three times for 5 min and counterstained with Hoechst 33342 (ab228551, Abcam) DNA-binding dye (0.2 μg/mL in distilled water) for 2 min. Finally, the samples were rinsed with distilled water and mounted using mounting medium (VitroGel^®^, 12–001, BioVitrum, Moscow, Russian Federation) and 0.13 mm coverslips (BB018024A1, Menzel Gläser, Braunschweig, Germany). The samples were assessed and photographed using a confocal laser scanning microscope (Fluoview FV10i, Olympus, Tokyo, Japan).

### 2.6. Enzyme-Linked Immunosorbent Assay (ELISA) of the Tear Samples

ELISA was done following the manufacturer protocols for each type of cytokines. Tear samples from each patient were collected in 0.25 mL Eppendorf-type test tubes, sealed, and stored in a low-temperature freezer (−40 °C). Quantitative assessment of the interleukins (IL) was performed by ELISA for IL-1b (A-8766), IL-2 (A8772), IL-4 (A-8754), IL-8 (A-8762) (Vector-Best, Russian Federation); IL-10 (BMS215-2) and TNF-α (BMS223-4) (Thermo Fisher Scientific, Waltham, MA, USA); and TGF-β (ab100647, Abcam, UK). The analysis of the samples was performed in one run. After storage, the samples were thawed at room temperature. The volume of all samples was adjusted to 200 µL with the manufacturer’s dilutions. The degree of dilution was used in the final calculations. A Multiskan GO Spectrophotometer and its proprietary software (Thermo Fisher Scientific, Waltham, MA, USA) were used for photometry and to calculate cytokine concentrations. Statistical analyses were performed using GraphPad Prism 7.

## 3. Results

After examination, four out of 34 (11.8%) patients were referred to the Oculoplastic Surgery Department for repair of the lid pathology (symblepharon, entropion, etc.). After that, 25 (73.2%) were scheduled for G-SLET, five (14.7%) for simultaneous anterior lamellar keratoplasty (ALK) and G-SLET, and four (11.8%) for para-limbal oral mucosa epithelium transplantation (PLOMET). From the entire cohort, 20 (58.8%) patients had definitive signs of LSCD and were referred for ocular surface reconstruction at the time of the initial examination, while for the other 14 (41.2%) patients, more comprehensive diagnostic methods were additionally used.

### 3.1. Comprehensive Diagnostics of Limbal Stem Cell Deficiency

#### 3.1.1. Etiology and Anamnesis

LSCD etiology may vary significantly from chemical and thermal burns to eye injuries of various etiologies, including conjunctival tumors, radiation exposure, infectious eye diseases, trachoma, prolonged soft contact lens use, chronic eyelid disease, surgical trauma of the limbal area, failed corneal grafts, and the use of systemic medications such as mitomycin C, 5-fluorouracil, chemotherapy, or immunotherapy.

Immune-mediated conditions are much more common in patients with bilateral LSCD, including Stevens-Johnson syndrome, Lyell syndrome, ocular cicatricial pemphigoid, congenital aniridia (PAX-6 syndrome), xeroderma pigmentosum, dyskeratosis congenital, autoimmune polyendocrine syndrome, lacrimo-auriculo-dento-digital (LADD) syndrome, Keratitis-ichthyosis-deafness (KID) syndrome, Ectrodactyly Ectodermal Dysplasia-Cleft Lip/Palate (EEC) syndrome, congenital epidermolysis bullosa, and idiopathic LSCD [1].

Careful and detailed analyses of patient history are extremely helpful. It can guide the decision-making process, especially for systemic comorbidities.

#### 3.1.2. Patient Complaints

The most common complaints are decreased visual acuity to light perception, chronic redness (hyperemia) of the conjunctiva, constant and sudden lacrimation and photophobia, blepharospasm and ocular pain (usually associated with recurrent corneal erosion), dry eye syndrome, and cosmetic defects [1]. It should also be considered that patients with severe LSCD may develop secondary glaucoma due to iridocorneal adhesions, retrocorneal membranes, and angle closure. These patients may complain of pain or periorbital fullness.

#### 3.1.3. Ocular Surface Biomicroscopy

First, attention should be paid to the involvement of the second eye in the pathological process, that is, the laterality of the lesion, which significantly influences treatment tactics. The most common signs of LSCD are corneal conjunctivalization at the periphery and/or the central part of the cornea, corneal opacities of varying intensity, superficial and deep corneal neovascularization, epithelial defects, and/or recurrent erosions. In the LSCD, it is not possible to clearly identify the architecture of the palisade Vogt at the limbus. In addition, there are various eyelid pathologies, such as symblepharon, lagophthalmos, and shortening of the conjunctival fornices.

The standard slit-lamp examination provides limited information regarding the state of the corneal epithelial layer. Therefore, it is vital to use low-molecular-weight fluorescein staining and examine eyes under cobalt blue light and a yellow filter, which allows assessment of the epithelial layer in more detail. Normally, the corneal epithelium is not stained with fluorescein, as the dye does not penetrate healthy corneal epithelial cells. In LSCD, the abnormal layer of conjunctival/metaplastic epithelium may present the patterns in the form of a curl or vortex (so-called “vortex pattern staining” or “whorl-like epitheliopathy”) [17] (Figure 1).

Two clinical examples illustrating the use of slit lamp examination with and without fluorescein staining to establish the diagnosis and guide the treatment tactics are presented in Figure 2 and Figure 3.

#### 3.1.4. Anterior Segment Optical Coherence Tomography

Anterior segment optical coherence tomography (AS-OCT) is a highly precise, non-invasive imaging modality that can be successfully used in patients with LSCD [25,26,27,28]. In our practice, we have experience with various machines but currently prefer to utilize the Avanti xR for AS-OCT examination and Angio-OCT modes using the AngioVue program and the SSADA algorithm.

The device allows accurate measurement of the thickness of the corneal epithelium and fibrovascular pannus, assesses the palisades of Vogt and limbal crypts, and visualizes the transition zone between the hypo-reflective corneal epithelium and hyperreflective conjunctival epithelium in the limbal area, as well as the intensity and depth of stromal opacities. Figure 4 demonstrates the visualization of the anterior chamber structures, which is also of great diagnostic importance because of the detection of multiple iridocorneal synechiae and retro-corneal membranes (having an inflammatory origin), which must be considered in further decision-making.

Corneal neovascularization in LSCD is a common clinical sign; thus, anterior segment OCT angiography (AS-OCTA) may be helpful, especially in examining the state of corneal neovascularization [29,30,31]. AS-OCTA allows non-contact targeting and observation of blood vessels in a certain area of the cornea, objectively monitoring the progression and regression of neovascularization before and after whatever treatment was applied [32]. In addition, using AS-OCTA, it is possible to identify small collateral vessels, which, in cases of diffuse and total corneal opacification, cannot be visualized by other non-invasive diagnostic methods [33].

Clinical examples illustrate the use of AS-OCT in examining healthy and LSCD-affected limbus areas and AS-OCTA in patients with LSCD and deep corneal neovascularization (Figure 4).

#### 3.1.5. In Vivo Confocal Microscopy

In vivo laser scanning confocal microscopy (IVCM) of the cornea is a non-invasive diagnostic tool for obtaining high-resolution images at various depths within the specimen [34,35,36,37,38]. In our practice, for comprehensive diagnosis of patients with LSCD, we used the IVCM machine HRT II with the Rostock Cornea Module (Heidelberg Engineering GmbH, Germany). This makes it possible to analyze the changes in the palisades of Vogt and various focal stromal areas, characterize cellular morphology (size, borders, nuclei, and cellular metaplasia), and evaluate the density and tortuosity of sub-basal corneal nerves. In patients with LSCD, we observed a decrease in the total sub-basal nerve density and the density of long nerves, which correlates with the severity of LSCD [39], a decrease in the density of basal epithelial cells, blurring, and the absence of a clear contour and shape (Figure 5).

#### 3.1.6. Impression Cytology and Immunofluorescent Staining (IC-IF)

The diagnosis of LSCD is not always obvious during a slit-lamp examination, even with fluorescein staining. Thus, impression cytology (IC) is a noninvasive diagnostic tool that can provide comprehensive information regarding the presence or absence of specific cellular keratins (K) and/or mucins [17]. A large number of different Ks are known, but K12 has strong specificity for the corneal epithelium [40,41], and at another point, K7, K13, and MUC5AC have been used for conjunctival cell detection [17,19,40,42]. Our experience demonstrated relatively similar results in detecting K7 and K13 as markers of conjunctival invasion. However, we prefer to use the K7 antibody and its selected clone so-called “OV-TL 12/30” due to its high specificity in detecting a clear border between corneal and conjunctival epithelium as it was demonstrated both in *post-mortem* and clinical samples [40]. Examples of K7 (clone OV-TL 12/30) and K12 expression in the clinical samples are shown in Figure 6.

#### 3.1.7. Pro-/and Anti-Inflammatory Cytokines in the Tear Samples

Long-term clear corneal transplant survival after keratoplasty is the result of a combination of several local (presence of a natural lens, absence of anterior synechiae, glaucoma, corneal bed vascularization, etc.) and systemic factors. Systemic factors include the activation of the recipient’s immune system, whose indicators (cytokines, antibodies, etc.) may be found in tears and blood samples. Several important factors are known to represent the activation of the immune response, including the ratio of individual populations of immunocompetent cells, the number of circulating immune complexes, levels of antibodies and cytokines, primarily interleukin-1β (IL-1β) and tumor necrosis factor-α (TNF-α) in blood serum and tears, as well as the presence of sensitization of immunocompetent cells to corneal antigens [43]. Thus, preoperative and postoperative monitoring of the recipient’s immune system indicators may provide useful and important information for guiding the management of each case.

We studied two groups of patients. The first group consisted of patients with total unilateral LSCD (*n* = 5), whereas the second group included patients with partial unilateral LSCD (*n* = 6) (Figure 7).

In the 1st group of patients (total LSCD), a high correlation between IL-8 and IL-4 expression was observed preoperatively, which was most probably due to the completion of tissue alteration and remodeling processes after the initial injury (corneal burn) (Appendix A). These interleukins behave as modulators and are not considered pro-inflammatory. By 6–12 months, the process of full integration of limbal auto-transplants into the corneal tunnels after G-SLET was observed clinically. During this period, the decrease in the inflammatory response correlated well with the cytokine profile. A significant decrease in IL-8 and IL-4 levels (r = 0.97) was observed in conjunction with a decrease in TGF-β activity, which correlated with the absence of IL-8 (r = 0.93) and IL-8 (r = 0.99). A significant decrease in the activity of other pro-inflammatory cytokines, such as TNF-α, in relation to TGF-β (r = 0.89) was also observed (Appendix A).

In Group 2 (partial LSCD), no inflammatory manifestations were observed before surgery. Under the influence of the modulating effect of IL-4, connective tissue remodeling was very likely due to the activation of transplanted limbal mesenchymal stem cells (MSCs). This was confirmed by laboratory tests indicating high levels of the cytokine TGF-β, which plays a role in stem cell differentiation, and highly correlates with IL-4, the latter having a proliferative activity (r = 0.68) (Appendix A). One year after the surgery, there was good correspondence between the clinical picture and cytokine profile indicators. The cytokine levels continued to decrease at the following ratios: IL-8 to IL-4 (r = 1.0), TGF-β to IL-8 (r = 0.91), and TGF-β to IL-4 (r = 0.9). The average correlation (r = 0.67) between the pro-inflammatory cytokines TNF-α and TGF-β may be due to the absence of minor signs of inflammation in the patients. These signs of inflammation may include chronic conjunctival hyperemia, epitheliopathy, blepharitis, and dry eye symptoms (Appendix A).

Our findings indicate that in Groups 1 and 2, pro-inflammatory cytokines (IL-4 and IL-8) may act as immunomodulators, playing a significant role in the inflammatory response, at least up to 6 months after surgery. The postoperative levels of fibroblast growth factor and TGF-β corresponded well with the clinical symptoms. The increase in interleukins and growth factors at 3 months postoperatively may be a result of surgical trauma followed by a decrease in their concentrations in both groups (Appendix A).

### 3.2. Surgical Interventions in Patients with Limbal Stem Cell Deficiency

To date, a broad range of techniques for corneal epithelium reconstruction have been suggested. For instance, in cases of unilateral LSCD, conjunctival-limbal (CLAU), auto-, and allo- kerato-limbal transplantations (KLAL) can be applied [44]. Surgical procedures can also utilize allogeneic cultured limbal epithelial cells from donor cadaver eyes [Schwab, 2000]. In 2012, simple limbal epithelial transplantation (SLET) was proposed [45]. The procedure involves corneal limbal graft harvesting from the upper limbus of a healthy eye, followed by fragmentation of the harvested tissue into 8–10 fragments and gluing fragments on top of the amniotic membrane glued to the LSCD-affected cornea after removing the fibrovascular pannus. Since appropriate fibrin glue was not available, Malyugin et al. proposed an original technique of G-SLET [46]. In this technique, the harvested limbal micro-fragments were fixed inside partial-thickness radial tunnels placed on the periphery of the cornea. Finally, one layer of cryopreserved human amniotic membrane (hAM) was sutured to the sclera outside the corneal limbus using a single running suture (vicryl 8-0).

#### 3.2.1. Limbal Stem Cell Transplantation in Patients with Unilateral LSCD

##### G-SLET in Patients with a Normal Corneal Thickness on the LSCD-Affected Eye

Since its introduction in 2020, we have made several modifications to the original G-SLET technique. Those were including corneal polishing with the diamond 0.5 mm burr (Algebrush II with 0.5 mm burr (Alger Company, Inc., Lago Vista, TX, USA) after pannus excision; using the original software to create the non-perforating corneal tunnels incisions with a femtosecond laser (LDV Z8, Ziemer, Switzerland); femtosecond laser excision and fragmentation of the limbal fragments from the donor’s eye; abandoning the use blepharonaro-rrhaphy and hAM; simultaneous anterior lamellar keratoplasty (ALK) with G-SLET in patients with extremely thin corneas.

A schematic representation of the principal steps of G-SLET in patients with normal or thick corneas is shown in Figure 8 and Figure 9.

##### Limbal Stem Cell Transplantation in Patients with Thin Cornea on the LSCD-Affected Eye

It has been reported that penetrating keratoplasty (PK) combined with limbal transplantation provides poor clinical results. Corneal graft opacification early postoperatively, together with recurrent conjunctivalization of the corneal surface, were documented in these cases [10,16]. In addition, PK combined with transplantation of cultured limbal epithelial stem cells, as a rule, does not lead to the desired anatomical success, that is, corneal re-epithelialization [47].

In cases where the cornea in the LSCD-affected eye is significantly thinner than normal (the stromal layer under the pannus is less than 250–300 microns according to AS-OCT), patients were scheduled for simultaneous anterior lamellar keratoplasty and G-SLET (Figure 9). This modification provided minimal risks of graft rejection, helped replenish the thickness of the cornea to physiological limits, and achieved better optical results in some cases.

#### 3.2.2. Corneal Re-Epithelization in Bilateral LSCD

In primary bilateral LSCD, LESC dysfunction is associated with systemic immune-mediated conditions (Stevens-Johnson syndrome, Lyell syndrome, rheumatic diseases) or genetic diseases (congenital aniridia) [1]. Both primary and secondary (burns, trauma, radiation exposure, etc.) bilateral LSCD are characterized by the complete absence of limbal stem cells, which are the source of healthy corneal epithelium. Therefore, these kind of patients are highly unlikely to benefit from any of the current surgical techniques of autologous limbal stem cell transplantation, as also from autologous CLET. Until recently, the treatment of bilateral LSCD was based on the implantation of a keratoprosthesis because of the high risk of corneal transplant failure [16]. However, the search for an effective way to re-epithelialize the cornea in bilateral LSCD is ongoing. Surgical techniques, such as alloLT, and technologies for cultured auto/allogeneic epithelial cell transplantation are currently being studied.

##### Allo-Limbal Transplantation

Allo-limbal transplantation (alloLT) is a surgical intervention for corneal epithelial reconstruction using a corneal limbal graft from a donor. AlloLT may be performed using a graft from a post-mortem donor or cadaver (c-), living donor of a living relative (lr-), or living donor of a non-relative (lnr-) [48]. According to the type of surgical intervention, alloLT is classified into autologous conjunctival limbal allograft (CLAL), kerato-limbal allograft (KLAL) [48], and allogeneic simple limbal epithelial transplantation (allo-SLET) [48]. The combined version of alloLT (lr-CLAL + c-KLAL) is referred to as the “Cincinnati procedure” [49]. The effectiveness of re-epithelialization with alloLT in patients with bilateral LSCD varied from 13 to 100% [16,50]. According to one study, alloLT in cases of post-eye burn LSCD from a living relative is more effective than using donors’ post-mortem grafts [50,51]. According to published data, re-epithelialization after alloLT is highly effective in LSCD as a complication of contact lens overuse but has relatively low success in patients with immune-mediated LSCD [50].

According to a meta-analysis, the most common complications of alloLT are recurrent/persistent corneal erosion (28.8%), ophthalmic hypertension (6.3%), corneal melting/perforation (5.6%), and infectious keratitis (4.2%) [52]. Among the complications of alloLT itself, there have been cases of transfer of malignant tumors from donor cadaveric material. The transmission of invasive lobular carcinoma [53] and conjunctival melanoma [54] has been documented.

##### Allogeneic Simple Limbal Epithelial Transplantation

Allo-SLET is a relatively new surgical approach in patients with bilateral LSCD. Its effectiveness reaches 87.5% with corneal limbal biopsy obtained from a living relative and 78.6% with biopsy obtained post-mortem. However, both approaches require systemic immunosuppression [55]. Various immunosuppressive protocols have been developed to downregulate the immune response during alloLT. They are mostly based on the systemic use of glucocorticosteroids, cytostatics, and immunosuppressants [52].

The most sophisticated approach for alloLT management was published by Holland [56]. AlloLT patients were examined and managed by a trained pharmacologist from a kidney transplantation team, according to a specific scheme [56]. According to a meta-analysis, the effectiveness of re-epithelialization with alloLT was significantly higher when 3 drugs is used in the protocol [52]. It is worth noting, that the age of patients receiving immunosuppression after alloLT was 38.4 ± 13.1 (mean ± SD) years [52].

##### Allogenic Cultivated Limbal Epithelial Stem Cell Transplantation

Allogenic cultivated limbal epithelial stem cell transplantation (allo-CLET) has been used in patients with bilateral LSCD for corneal reconstruction since 2000 [57]. The efficiency of re-epithelialization in allo-CLETs varied from 50% to 71.4% [13,16,52]. The main complications of allo-CLET are corneal erosion, ophthalmic hypertension, corneal melting and/or perforation, and infectious keratitis [52].

Several studies have reported poor outcomes of corneal re-epithelialization using the allo-CLET procedure in patients with aniridia-associated keratopathy (AAK). According to a previous study (13 patients, 14 eyes), the positive effect of the procedure decreased over time, and at the follow-up period of 40 months, healthy corneal epithelium was present only in 25% of recipients [58]. According to another study, the primary positive effect (80%) of allo-CLET with HLA donor matching (6 patients, 6 eyes) progressively failed over time, and AAK regressed to the preoperative level in most of the patients during the follow-up period of 53.6 months [59].

Systemic immunosuppression, as necessary support for allogeneic transplantation of cultured cells, was used in all allo-CLET cases. However, these protocols differ significantly in terms of the number of drugs used, their combinations, and dosing regimens. The most commonly used drugs are corticosteroids, tacrolimus, cyclosporine, mycophenolate mofetil, and azathioprine [52].

##### Cultivated Oral Mucosal Epithelial Transplantation

The first clinical results of autologous cultured oral mucosal epithelial transplantation (COMET) for corneal re-epithelialization in patients with bilateral LSCD were published in 2004 [60]. According to the reviews, the anatomical success of COMET varies from 66.7 to 81.5%, while visual acuity improvement was registered in 68 to 79.0% of cases [13,61,62,63,64] Penetrating keratoplasty in successful re-epithelization after COMET was associated with a high overall corneal donor graft survival probability of up to 92.9% [65].

##### Transplantation of the Epithelial Cells Generated from Induced Pluripotent Stem Cells

Human corneal epithelial cells from induced pluripotent stem cells (iPSCs) were first obtained by Hayashi et al. in 2016 [66]. However, this protocol involves very high costs and low efficiency because other types of cells are formed in the colonies, such as retinal pigment and the lens epithelium. Later, other researchers improved the protocols for cellular reprogramming and cultivation to increase the efficiency of obtaining human corneal epithelial cell colonies [67].

The application of corneal epithelial cells from iPSCs for re-epithelialization in bilateral LSCD has advantages owing to its potential autologous characteristics and tissue specificity. However, the technology used for obtaining cells is very complicated. In addition, the costs of its implementation is more expensive in comparison with the cultivation of somatic cells (LESC, oral mucosal epithelium). In addition, highly qualified personnel and a complex quality control and standardization system are required.

The first clinical trial on the transplantation of iPSC-derived corneal epithelium is currently underway in Japan (JPRN-UMIN000036539, May 2019) [68]. The research team planned to transplant these cells into four patients with LSCD. However, the protocol for the clinical study indicated that an allogeneic source would be used [68], which likely requires systemic immunosuppression in patients.

##### Simple Oral Mucosal Epithelium Transplantation

An alternative and emerging surgical technique for corneal re-epithelialization in bilateral LSCD is autologous simple oral mucosal epithelium transplantation (SOMET) [69,70,71,72]. SOMET requires direct grafting of the oral mucosal epithelium onto the corneal periphery after pannus release and amniotic membrane overlay with the help of fibrin glue (Figure 10). For these, a strip of the labial or buccal mucosa is trimmed off from the underlying *substantia propria*, chopped into 13–15 mini-grafts, and glued on top of the amniotic membrane at the corneal periphery [70]. The advantages of this technique are relative simplicity, low cost, and, of great importance, that there is no need for a donor corneal limbus graft and/or systemic immunosuppression.

SOMET has the same rationality for use as COMET because the labial or buccal mucosa in humans is covered with stratified squamous, non-keratinized epithelium. SOMET has also been tested in animal models, and LSCD-induced corneas were successfully re-epithelized after SOMET [69,73].

##### Paralimbal Oral Mucosal Epithelium Transplantation

Paralimbal oral mucosal epithelium transplantation (PLOMET) is another emerging surgical technique for corneal re-epithelialization in patients with bilateral LSCD, based on the same rationale as COMET and SOMET. In PLOMET, a long strip of the labial mucosal epithelium was harvested and attached to the sclera 3 mm away from the corneal limbus after pannus release (Figure 11). Variations of the technique include circumferential trephination of the lip mucosa [74], application of an amniotic membrane as an overlay, and fibrin glue [75].

The concept of PLOMET has been experimentally studied in an acellular human corneal model. For this, a labial mucosal epithelium strip was sutured at the corneal periphery and cultured. 21 days after the cornea was covered with oral mucosal epithelium, K4, K13, and K19 were expressed [76]. Patients are currently being recruited for the study to evaluate the effectiveness of PLOMET (https://clinicaltrials.gov/ct2/show/NCT04995926, accessed 29 December 2022). Also, it is to be mentioned, that the optimal corneal re-epithelization technique in bilateral LSCD caused by systemic immune-associated or genetic diseases is yet to be defined.

## 4. Discussion

Reconstruction of the corneal epithelium is the primary treatment for LSCD. Until recently, these patients had either unsuccessful keratoplasty or keratoprosthesis. In our practice, we examined multiple patients who had undergone PKP more than once, had prolonged courses of pharmacological treatment, were wearing therapeutic contact lenses, covered the cornea with hAM, and received various medications to suppress angiogenesis. All of the above-mentioned approaches may be classified as symptomatic treatment modalities with limited and temporary effects.

At the core of the treatment strategy, the first line of treatment is the correct diagnosis. It should be mentioned that in patients with LSCD, the diagnosis is not always straightforward or obvious. Slit lamp biomicroscopy with fluorescein staining has always been the mainstream method. However, laboratory diagnostics play a significant role in identifying this condition. Here, we propose a diagnostics and treatment algorithm for LSCD (Figure 12).

LSCD—limbal stem cells deficiency; OCT—optical coherence tomography; AS-OCTA anterior segment optical coherence tomography angiography; ELISA—enzyme-linked immunosorbent assay; IVCM—in vivo confocal microscopy; IC—impression cytology; CK—cytokeratins; CLAL—conjunctival limbal allograft; KLAL—keratolimbal allograft; SLET- simple limbal epithelial transplantation; G-SLET—glueless simple limbal epithelial transplantation; CLET—cultivated limbal epithelial transplantation; CLAU—conjunctival limbal autograft; KLAU—keratolimbal autograft; ALK + G-SLET—anterior lamellar keratoplasty + glueless simple limbal epithelial transplantation; FS-G-SLET—femtosecond laser-assisted glueless simple limbal epithelial transplantation (FS laser is used to create cornel tunnels); FS-limb + G-SLET—femtosecond laser-assisted glueless simple limbal epithelial transplantation (FS laser is used to create corneal tunnels and harvest the limbal fragment from the healthy eye); PLOMET—paralimbal oral mucosa epithelial transplantation; SOMET—simple oral mucosa epithelial transplantation; COMET—cultivated oral mucosa epithelial transplantation.

At the first appointment, we ask about the main complaints such as decreased visual acuity, lacrimation, photophobia, and dry eye, collecting medical history, paying special attention to the facts of chemical or thermal burns, keratitis, failed repeated keratoplasties, and the use of contact lenses. We then examined the ocular surface using a slit lamp and performed fluorescein staining. OCT is the first and one of the most important instrumental diagnostic methods in our algorithm, which allows us to investigate corneal thickness, perform epithelial mapping and study the structure of the Vogt palisades. If OCT signs were controversial and did not confirm the diagnosis, we performed in vivo confocal microscopy (IVCM) and anterior-segment optical coherence tomography angiography (AS-OCTA) to assess cellular structures and corneal blood vessels. Using all the above-mentioned diagnostic methods, we can establish a final diagnosis in many cases. However, if doubtful, impression cytology should be performed to evaluate specific keratins. Cytokeratin 7 (and/or) cytokeratin 19 can be used to confirm the diagnosis of LSCD. Afterward, we chose an adequate surgical method to treat the unilateral or bilateral LSCD.

It is worth concluding that in most patients with LSCD, visual acuity decreases significantly, and subsequent visual and functional rehabilitation may take a considerable amount of time. It depends on the collateral damage to the eyelids, lid fornices, corneal stroma, and intraocular structures. Formation of symblepharon, eyelid deformations, and lagophthalmos significantly affect the outcomes of corneal surface reconstruction. Also, it should be mentioned that many of these patients become disabled, and their quality of life is significantly reduced, not to mention psychological trauma.

Adequate surgical methods for the treatment of unilateral and bilateral LSCD are debatable. In bilateral LSCD, there are three main directions of treatment: allogeneic and autologous transplantation of stem cells and keratoprosthesis. We consider autologous transplantation to be very promising, as it does not require prolonged immunosuppression. Therefore, we propose paralimbal oral mucosal epithelium transplantation (PLOMET) as a method of using the strip of labial mucosal epithelium and its suture fixation in the paralimbal area. So far, we have shown promising results for PLOMET in patients without systemic immune-associated or genetic diseases.

In this paper, various techniques of surgical LSCD treatment are discussed and proposed as a standardized approach allowing for better and higher clinical outcomes. All of the technologies mentioned are not without their drawbacks. In some cases, only a small group of patients has been studied within a relatively short period. In others, not all clinical and functional outcomes are well-defined. Many surgical approaches lack standardization, are time-consuming, and require surgical dexterity and special equipment. Nevertheless, despite all the challenges, many new technologies help to improve the clinical results and make this surgery very encouraging for the surgeon and patient alike.

## 5. Conclusions

Based on our experience, we were able to identify the key methods and points in LSCD diagnostics, allowing us to build an algorithm that helps guide the surgeon in the selection of an appropriate surgical procedure.

## Figures and Tables

**Figure 1 diagnostics-13-00199-f001:**
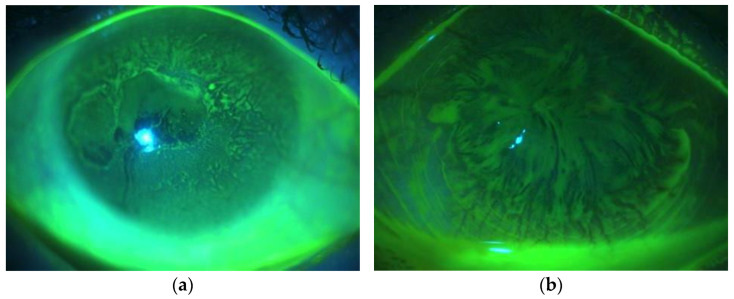
Slit-lamp biomicroscopy images pattern of abnormal fluorescent staining. (**a**) the central area of normal corneal epithelium is surrounded by the epithelium with abnormal staining (partial LSCD), and (**b**) “whorl-like epitheliopathy” (total LSCD).

**Figure 2 diagnostics-13-00199-f002:**
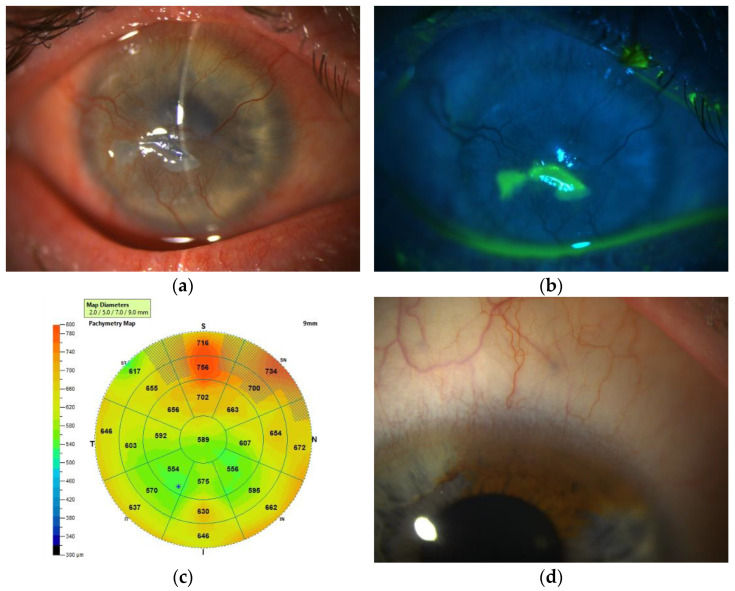
Slit-lamp biomicroscopy with and without fluorescent staining in a patient with unilateral LSCD. (**a**,**b**) Slit-lamp biomicroscopy images of the affected right eye (**a**) and healthy corneal limbus of the left eye (**d**). On the right eye (**a**) corneal epithelium is edematous and opaque, the palisades of Vogt are absent, intensive superficial corneal neovascularization, and central corneal epithelial erosion can be seen. (**b**) Slit-lamp biomicroscopy image and fluorescent staining with cobalt light and a yellow filter. Central corneal epithelial erosion is visible. (**c**) AS-OCT of the LSCD-affected eye. 9 mm corneal pachymetry map. The final diagnosis is RE—total unilateral LSCD, persistent epithelial defect. Normal corneal thickness under the pannus. The patient was scheduled for G-SLET on that eye.

**Figure 3 diagnostics-13-00199-f003:**
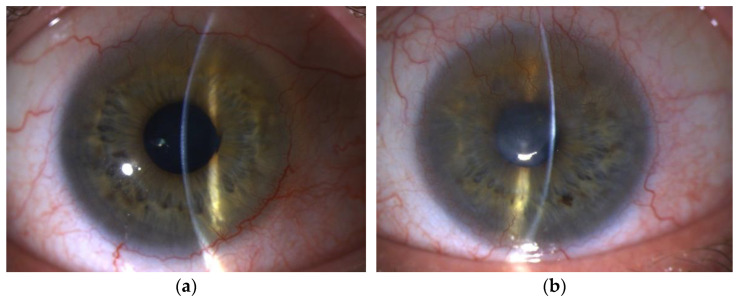
Slit-lamp biomicroscopy with and without fluorescent staining in a patient with bilateral LSCD. (**a**,**b**) Slit-lamp biomicroscopy images. Superficial peripheral corneal neovascularization on the right (**a**) and left (**b**) corneas. On the right eye (**a**) corneal epithelium is transparent and stable, but the palisades of Vogt are absent. On the left eye (**b**) conjunctival invasion from the upper limbus is reaching the optical center, the epithelium is edematous and opaque, and the palisades of Vogt are absent. (**c**,**d**) Slit-lamp biomicroscopy images and fluorescent staining with cobalt light and a yellow filter. The furrow staining pattern of the bulbar conjunctiva on the right (**c**) and left (**d**) eyes. Definitive line of conjunctival invasion of the left cornea, part of the corneal epithelium is retained (**d**). Diagnosis: BE—contact-lens induced bilateral partial LSCD, normal thickness of the corneal stroma in both eyes. The patient was enrolled in the clinical trial of paralimbal oral mucosa epithelium transplantation (PLOMET) on the left eye (NCT04995926).

**Figure 4 diagnostics-13-00199-f004:**
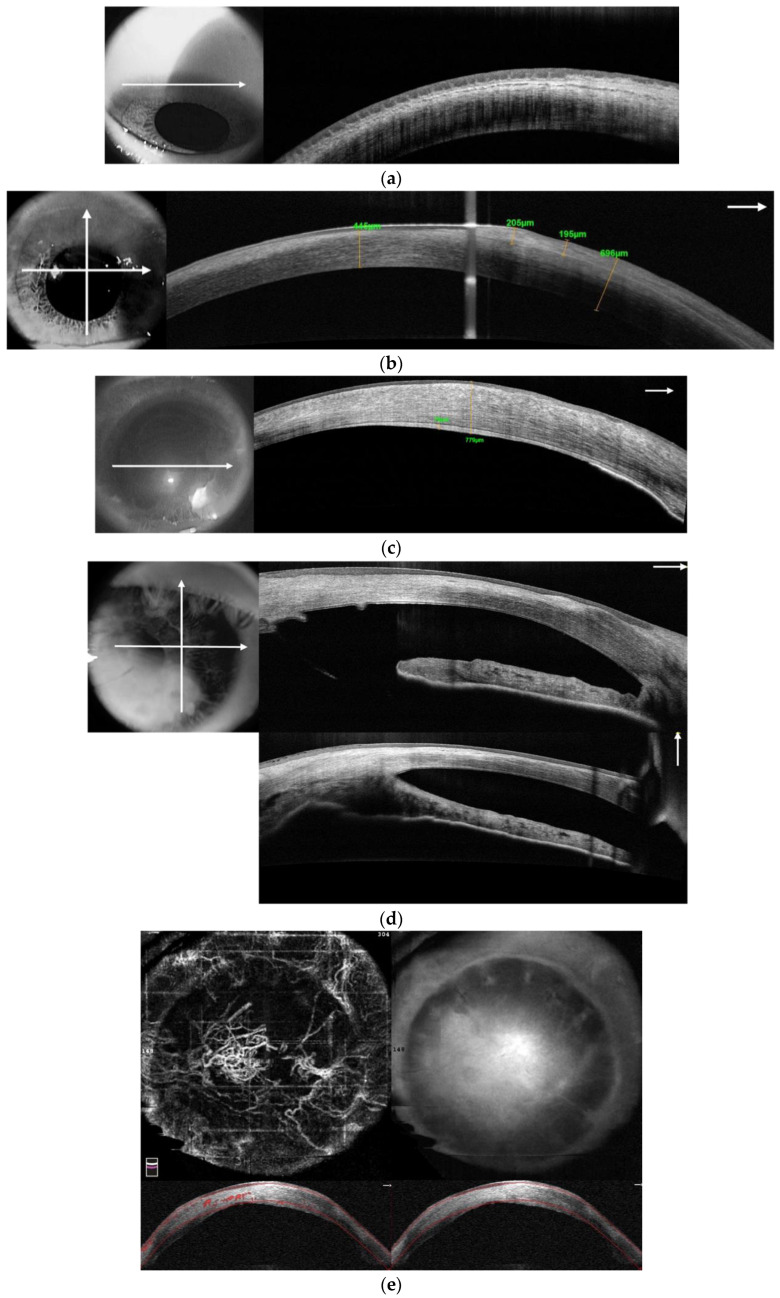
AS-OCT of the cornea. (**a**) Composite image of the upper corneal limbus of the healthy eye of the patient with unilateral LSCD. The sagittal cross-sectional OCT images demonstrate the typical crest-like structures of the palisades of Vogt. (**b**) Composite OCT image of the cornea of LSCD-affected eye. A horizontal cross-section demonstrated a clear boundary between the corneal epithelium and conjunctival invasion having a thickness of 200 microns. (**c**) Composite OCT image of the cornea of the LSCD-affected eye. A well-defined hyperreflective retro-corneal membrane can be visualized on the horizontal cross-section. (**d**) Composite OCT image of the cornea after G-SLET procedure. An iris-corneal synechiae in the superior (horizontal cross-section) and inferior (sagittal cross-section) parts of the anterior chamber are visualized on the LSCD-affected eye. (**e**) Composite image of AS-OCTA of the LSCD-affected eye with previously failed PKP. Superficial and deep corneal neovascularization (red dots) can be visualized.

**Figure 5 diagnostics-13-00199-f005:**
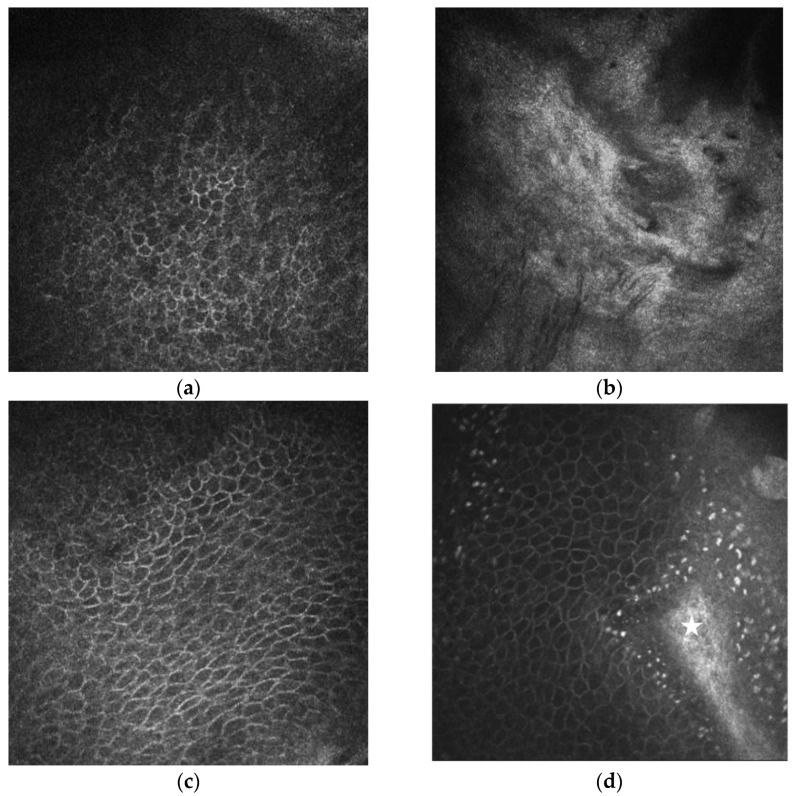
In vivo confocal microscopy of the patient with partial unilateral LSCD. (**a**) Confocal image of epithelial basal cells at the central cornea of the LSCD-affected eye. Wing cells are morphologically abnormal, intercellular borders are not well-defined. (**b**) Confocal image of limbal epithelium and the palisades of Vogt of the LSCD-affected eye. Absent are the palisades, and the structures of the limbus are severely damaged. (**c**) Confocal image of epithelial basal cells at the central cornea 10 months after G-SLET. Normal appearance of the wing cells having dark cytoplasm, well-defined and bright borders, and no visible nuclei. (**d**) Limbal micro-transplant (white asterisk) located inside the corneal tunnel and surrounded by the wing cells with well-defined bright borders in patient 6 months after G-SLET. (**a**–**d**) Image size 400 × 400 microns.

**Figure 6 diagnostics-13-00199-f006:**
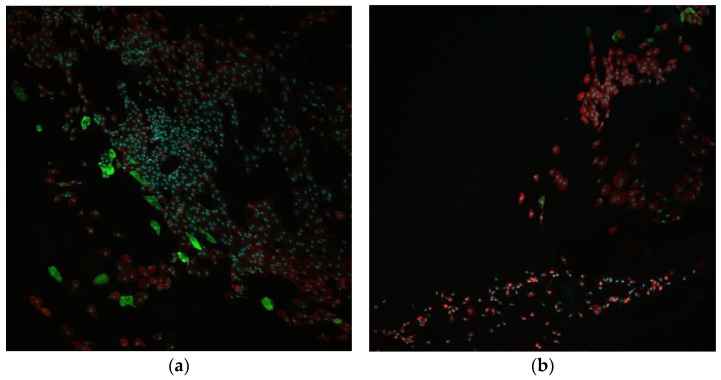
Impression cytology and immunofluorescent staining of the samples obtained from the LSCD-affected eye. (**a**) Central corneal sample. Nuclei are many, well stained with many K7 positive cells (red) with moderate expression (++), and a small number of K-12 positive cells (green) with strong expression (+++). (**b**) Corneal periphery sample. Nuclei are many, well stained; many K7 positive cells (red) with strong expression (+++), and few K12 positive cells (green) with strong expression (+++). Anti-K7 (OV-TL 12/30) (1:100, ab216016, mouse monoclonal, Abcam, UK); anti-K12 (1:100, ab185627, rabbit monoclonal, Abcam, UK). Confocal laser scanning microscopy, ethanol-fixed samples, immunocytochemistry, and immunofluorescence: green—Alexa Fluor^®^488 (1:250, ab150077, goat anti-rabbit, Abcam), red—Alexa Fluor^®^594 (1:250, ab150116, goat anti-mouse, Abcam), blue—Hoechst 33342 (ab228551, Abcam).

**Figure 7 diagnostics-13-00199-f007:**
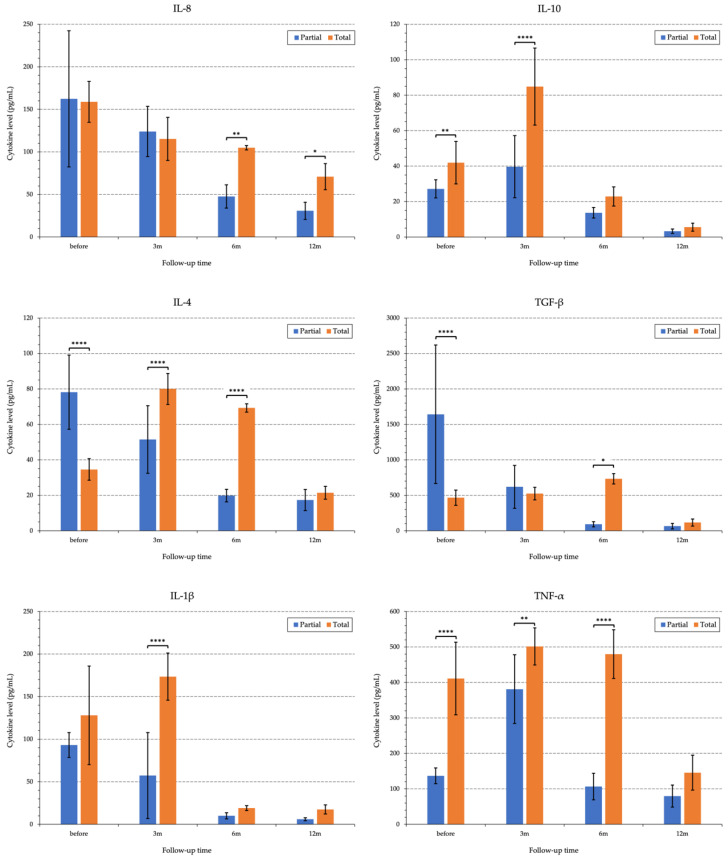
Bar chart of the cytokines levels in tear samples pre-operatively and after GSLET in patients with unilateral LSCD. Columns are presented as mean ± standard deviation. *p*-values: *—*p* < 0.05, **—*p* < 0.01, ****—*p* < 0.0001. LSCD—limbal stem cell deficiency, m—month, IL—interleukin, TGF—transforming growth factor, TNF—tumor necrosis factor, m—month, pg/mL—picograms per milliliter.

**Figure 8 diagnostics-13-00199-f008:**
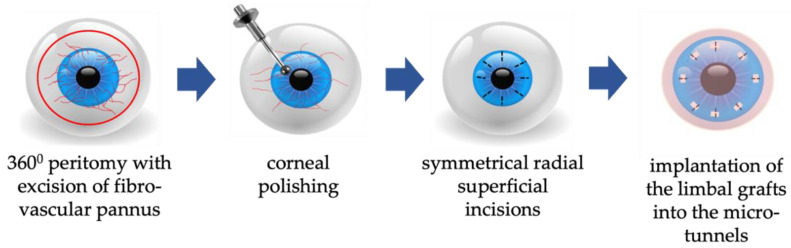
Schematic representation of the main steps of G-SLET in unilateral LSCD. After 360 degrees of peritomy and excision of the fibrovascular pannus, cornea polishing is performed with the diamond-dusted burr. 8 symmetrical radial non-perforating incisions are made for implantation of mini-limbal grafts taken from the upper corneal limbus of the healthy eye. Surgery is finished with suturing an amniotic membrane overlay.

**Figure 9 diagnostics-13-00199-f009:**
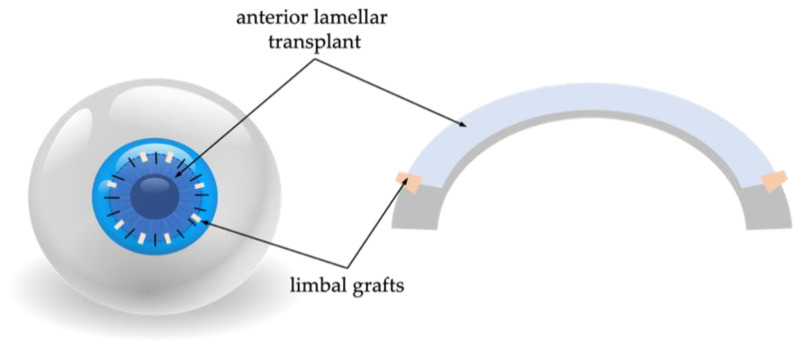
Schematic representation of the main steps of simultaneous anterior lamellar keratoplasty (ALK) and G-SLET in patients with thin corneas. After 360 degrees peritomy ALK is performed. Single sutures (10-0 nylon) are used for donor corneal transplant fixation. Limbal mini-grafts are placed in the donor-recipient interface.

**Figure 10 diagnostics-13-00199-f010:**
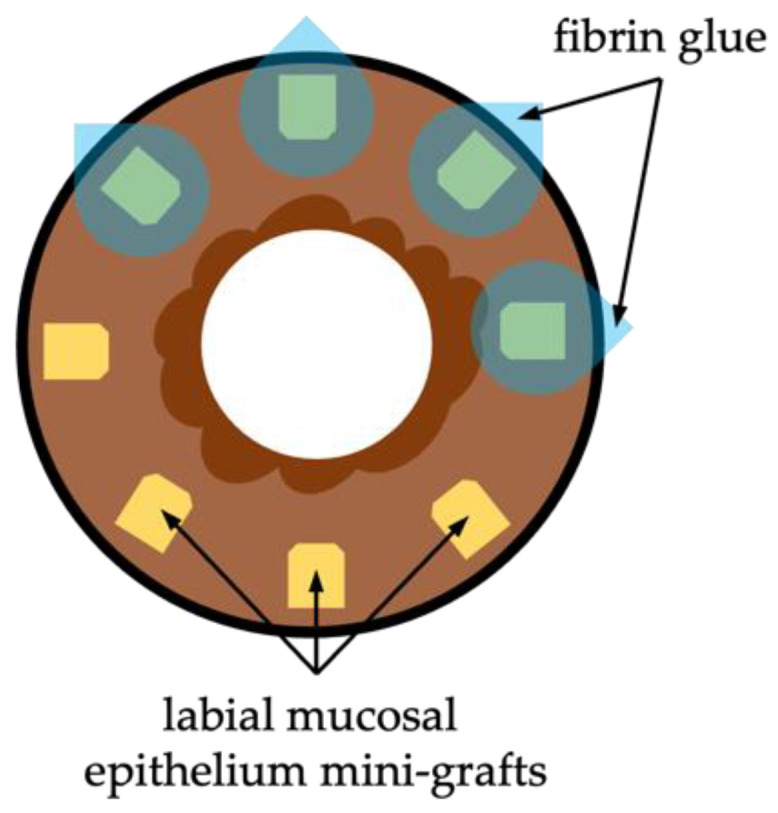
Schematic representation of simple oral mucosal epithelium transplantation (SOMET) for corneal re-epithelization in bilateral LSCD. Labial mucosal epithelium graft is divided into mini-grafts and attached to the affected cornea with the fibrin glue.

**Figure 11 diagnostics-13-00199-f011:**
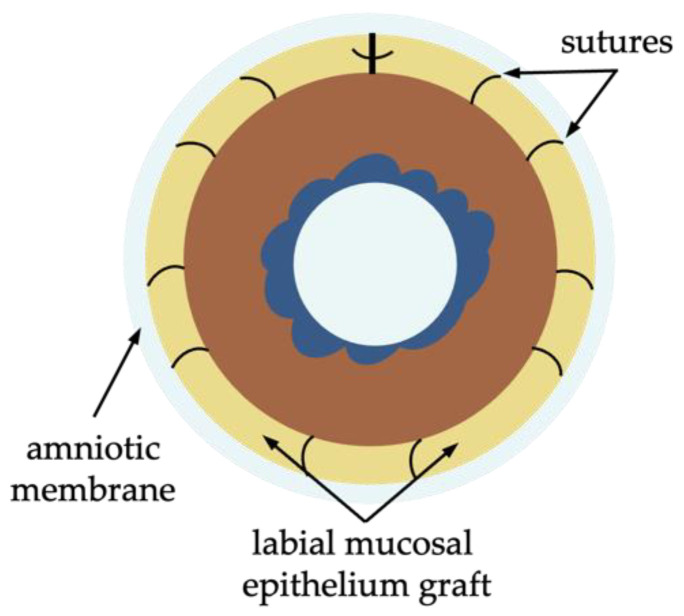
Schematic representation of paralimbal oral mucosal epithelium transplantation (PLOMET) for use in bilateral LSCD. The labial mucosal epithelium graft is sutured 2–3 mm away from the corneal limbus surrounding the affected cornea.

**Figure 12 diagnostics-13-00199-f012:**
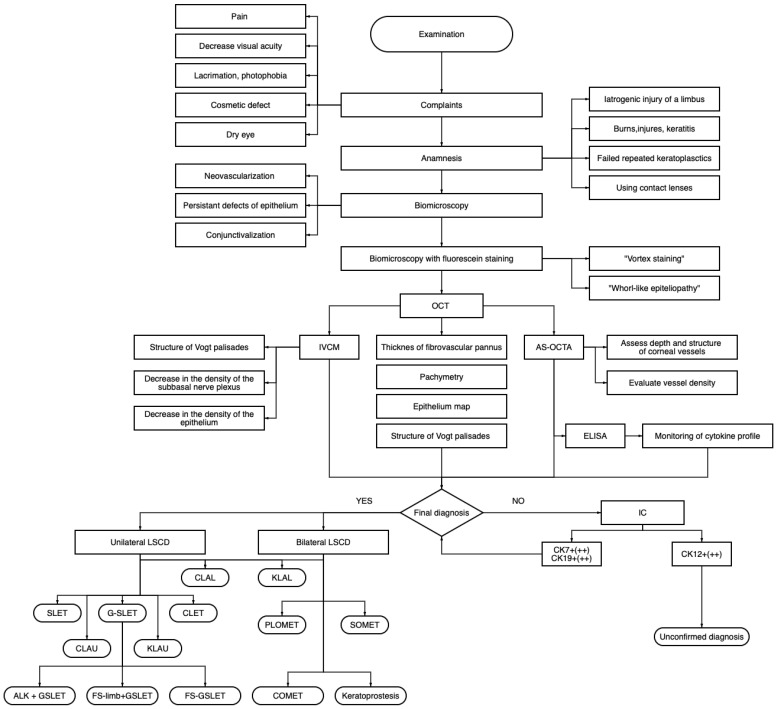
LSCD diagnostics and surgical treatment algorithm.

## Data Availability

Data supporting reported results can be available upon request.

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
