# Peer review of "Diagnostic Algorithm for Surgical Management of Limbal Stem Cell Deficiency"

_diagnostics, 2023, doi:10.3390/diagnostics13020199_

Round 1

Reviewer 1 Report

This paper investigated various diagnostic methods used in LSCD patients and identified several critical decision-making parameters. However, their paper need further experiments or evidences to support their conclusion.

Specific major points:

1.The detailed methods of Fluorescein stain, AS-OCT, and IVCM are needed in their Methods.

2.Do not combine all cytokines in one graph. Plz, show the statistical changes of each cytokine during f/u time on each subfigure in Fig.7 and Fig.8.

3.The authors need to analyze their data of F-stain, AS-OCT, impression cytology, and IVCM according to partial versus total LSCD to identify critical decision-making parameters of each diagnostic method. It is very important to decide the diagnosis and managements (especially surgical methods) of LSCD in Fig.13 

Author Response

Respected Sir/Madam, We appreciate your time and efforts spent reviewing our article. You will find the answers to your questions in the attached file. The article was amended according to your suggestions. Sincerely yours, Boris Malyugin, Corresponding author. 

Reviewer 2 Report

How many different methods were used to perform optical coherence tomography on the front of the eye?
How many distinct procedures for detecting LSCD exist that are both reliable and repeatable?
What kind of medical therapy did the treating doctors have in mind for a patient who was diagnosed with a unilateral long-segment deficit?

When discussing paralimbal oral mucosa epithelial transplantation, what was the intended message?
How many different changes can happen to the balance of the corneal epithelium when the cornea's homeostasis is upset?
Which of the following has a stable corneal epithelium?
How can an LSCD eye avoid injury to the corneal epithelium?
What does the term "complete diagnosis" mean in the context of limbal stem cell deficits?
How many decisions does LSCD cease to assist with?
Can you tell me how far the hurt cornea is from the grafted mucosal epithelium on the labial surface?
Can behavioural changes effectively treat long-term sleep apnea syndrome?
When would keratoplasty or keratoprosthesis have been attempted if it had previously failed?
Which of the following categories best summarises the various strategies that have been presented thus far?
How can the presence of LSCD be determined?
In what percentage of cases did OCT fail to produce a definitive diagnosis?
How do we know which steps of a diagnosis to take first?

Introduction needs to explain the main contributions of the work more clearly.

The novelty of this paper is not apparent. The difference between present action and previous Works should be highlighted.

The author should depict the flow graph to illustrate the need for the proposed approach.

The significant trends of the simulation results should show.

Comparison with recent studies and methods would be appreciated.

Introduction section can add the issues in the current work context and how proposed algorithms/approaches can overcome this.

Literature review techniques have to be strengthened by including the current system's issues and how the author proposes to overcome the same.

Clarify the finding Error rate and accuracy in the performance analysis section.

It is suggested to add the chart for the given process with a description.

The mapping process for the proposed technique should be discussed in detail.

Conclusion should state scope for future work.

Author Response

(The authors gave the same response as above.)

Round 2

Reviewer 1 Report

This paper need to be marked and highlighted the revised parts as different color and they also need to address and update according to below comments. 

Specific major points:

1.The detailed methods of Fluorescein stain, AS-OCT, and IVCM are needed in their Methods.

2.Do not combine all cytokines in one graph. Plz, show the statistical changes of each cytokine during f/u time on each subfigure in Fig.7 and Fig.8.

3.The authors need to analyze their data of F-stain, AS-OCT, impression cytology, and IVCM according to partial versus total LSCD to identify critical decision-making parameters of each diagnostic method. It is very important to decide the diagnosis and managements (especially surgical methods) of LSCD in Fig.13 

Author Response

Dear Sir/Madam,

We are sending you the revised manuscript with changes highlighted.

Respectfully yours,

Boris Malyugin

Round 3

Reviewer 1 Report

I accepted the manuscript.